# Current Global Health Impact Assessment Practice

**DOI:** 10.3390/ijerph17092988

**Published:** 2020-04-25

**Authors:** Mirko S. Winkler, Peter Furu, Francesca Viliani, Ben Cave, Mark Divall, Geetha Ramesh, Ben Harris-Roxas, Astrid M. Knoblauch

**Affiliations:** 1Swiss Tropical and Public Health Institute, P.O. Box, CH-4002 Basel, Switzerland; 2University of Basel, P.O. Box, CH-4001 Basel, Switzerland; 3Global Health Section, Department of Public Health, University of Copenhagen, P.O. Box 2099, 1014 Copenhagen K, Denmark; 4International SOS, Vesterbrogade 149, 1620 København V Copenhagen, Denmark; 5BCA Insight Ltd., 5-7 St Pauls Street, Gresham House, Leeds LS1 2JG, UK; 6Department of Geography and Planning, University of Liverpool, 74 Bedford Street South, Liverpool L69 7ZQ, UK; 7Centre for Primary Health Care and Equity, University of New South Wales, Sydney 2052, Australia; 8Shape Consulting, P.O. Box 602, St Peter Port GY1, Guernsey, UK; 9Advisian, 151 Canada Olympic Rd, Calgary, AB T3B 6B7, Canada; 10Population and Community Health, South Eastern Sydney Local Health District, 301 Forbes St, Darlinghurst 2010, Australia

**Keywords:** determinants of health, health impact assessment, planetary health, sustainable development goals

## Abstract

Health impact assessment (HIA) practice has expanded across the world, since it was established more than two decades ago. This paper presents a snapshot of current global HIA practice based on the findings of an online questionnaire survey. HIA practitioners from all world regions were invited to participate. A total of 122 HIA practitioners from 29 countries completed the survey, following a broad international outreach effort. The large variety in the types of HIAs conducted, and the application of HIA in various fields reported by respondents, demonstrates that HIA practice has evolved over the past two decades. Although differences in the use of HIA were reported across world regions, an overall increasing trend in global HIA practice can be observed. In order to sustain this upward trend, efforts are needed to address the main barriers in the utilisation of HIA. The establishment of new national and international HIA teaching and training offerings seems to be an obvious strategy to pursue along with the strengthening of policies and legal frameworks that specify the circumstances, under which HIA is required, and to what extent.

## 1. Introduction

Increasing recognition of the manifold implications of human-induced policies, plans, programmes, or projects on the environment led to the development of the environmental impact assessment (EIA) approach as an important component of environmental management in the 1960s [1,2]. With the enactment of a revolutionary piece of legislation, namely the National Environmental Policy Act, promulgated in 1969, the United States of America became the first country that established a comprehensive legal basis for EIA. While most countries followed the example of the United States of America [3], it took three decades until the notion of health impact assessment (HIA) was coined as a separate form of impact assessment with a specific focus on human health [4,5].

Since then, HIA practice has grown and diversified, with practitioners from the United Kingdom, the United States of America, Canada, Australia, and other European countries pioneering this evolving public health discipline [6]. Birley’s first book “HIA of development projects” (1995) [7], the 1999 Gothenburg consensus paper on HIA [8], the first book describing HIA concepts, theory, techniques, and applications by Kemm and colleagues (2003) [9], the “HIA best practice principles” of the International Association for Impact Assessment (IAIA; 2006) [10], Birley’s (2011) book on “HIA: principles and practices” [11], and the second HIA book by Kemm and colleagues (2012) [12] can be considered milestones in the international development and definition of the HIA approach. The overview of the international historical developments in HIA, published by Harris-Roxas and colleagues in 2012 [13], shows that the HIA approach has matured, diversified and expanded to an increasing number of countries globally, with applications in both, the public and private sector [14].

In this paper, we present a snapshot of current global HIA practice, based on a survey targeting HIA practitioners from all over the world. The main research questions of the study were: What are the characteristics (e.g., country of origin, countries of practice, years of experience, types of HIA conducted, fields of HIA applications) of the current cohort of HIA practitioners? What trends in HIA use do HIA practitioners perceive? What are the promoters and barriers of current HIA practice? What are the primary resource documents and capacity building efforts guiding HIA practice? What are the specific elements that may need to be considered in future international HIA practice guidance documents? This paper is part of an ongoing effort in updating the “IAIA HIA international best practice principles” [10], which aims to take into account the experiences, views and needs of HIA practitioners across all world regions.

## 2. Materials and Methods

A descriptive study was performed using an online questionnaire (SurveyMonkey^®^). For the purposes of the questionnaire, HIA practice was defined for participants as comprising a stand-alone HIA, and any other form of health in impact assessment, such as health in (EIA) or environmental, social, and health impact assessment (ESHIA). The questionnaire tool included a mix of both closed and open-ended questions around the following topics:Background: country of current host institution/employer and type of employer/host institution (e.g., non-profit organisation, company, research institutions, governmental authority).HIA practice: years of experience, other HIA functions in addition to being a practitioner, primary regions of HIA practice (using World Bank categories [15], with the exception that “Europe and Central Asia” were considered two separate regions), HIA application fields (policies, plans, programmes, projects, strategies), types of health assessments conducted (e.g., HIA, health in EIA), and decision-maker of HIAs conducted.Perceived trends in HIA use (decreasing, stagnant, increasing) in regions with HIA experience.Perceived drivers and barriers for HIA in countries with HIA experience.HIA teaching and training (received and facilitated).Preferred source documents for guiding own HIA practice.General questions on the “IAIA HIA international best practice principles” [10] (e.g., reasons for having consulted the paper in the past, need for revisions).Specific technical questions on the “IAIA HIA international best practice principles” [10] (not reported in detail here as beyond the scope of the paper).

Target participants for the survey were self-identifying HIA practitioners from across the world. The online survey was accessible from December 2018 to July 2019. Participants identifying information were not collected.

In order to launch the survey, an invitation message, explaining the research and containing the questionnaire link, was sent by e-mail to the “HIA network” of the authors (personal contacts and HIA focal points of selected institutions) and disseminated over social networks (LinkedIn HIA group, Twitter @HIAblog, IAIA Connect). At the beginning of the questionnaire, an introductory text was presented and the respondents gave their consent to participate. The introduction also listed the inclusion criteria that aimed at restricting participation in the survey to HIA practitioners, defined as having been involved in conducting HIA or implementing the outcomes of HIA. The exclusion criterion was not having been directly involved in HIA of a policy, plan, programme, project, or strategy. Simple screening or high-level review of assessments did not qualify participation in the survey.

In addition to the dissemination of the questionnaire through e-mail and social media, the HIA practice survey was advertised at the 39th Annual Conference of the IAIA (29 April–2 May 2019) in Brisbane, Australia. In a dedicated session, with the majority of the respondents being IAIA Health Section members, an update on the present study was given and strategies to invite other HIA practitioners from across all world regions were discussed. This led to further snowball dissemination of survey invitations via e-mail through the personal networks of the session participants (approximately 20). In addition, a survey invitation e-mail was sent to all IAIA Health Section members (mailing list with >200 members).

The collected data were exported from SurveyMonkey^®^ to Microsoft Excel for subsequent data cleaning and analysis. Descriptive quantitative analysis was performed in STATA statistical software version 15 (StataCorp LP; College Station, TX, USA). Data from open-ended questions were reviewed and grouped into meaningful categories where necessary. Since the questions asked in the online questionnaire were answered by varying numbers of respondents, the nominator (number of answers per answer category [x]), denominator (total number of responses received per question [y]) and percentage (% [x/y]) are specified for all the results presented.

## 3. Results

### 3.1. Study Participants

In total, 122 HIA practitioners from 29 countries completed the survey, providing a good global perspective (Figure 1a).

Just over half of the survey respondents (51.6% [63/122]) were based in the European region, followed by North America and East Asia and Pacific (15.6% each [19/122]) (Figure 1b). Most respondents worked for private (for-profit) organisations (36.1% [44/122]), research or higher education institutions (32.0% [39/122]), and governmental authorities (22.1% [27/122]). Only few participating HIA practitioners were from non-profit organisations (4.9% [6/122]) or identified themselves as self-employed (1.6% [2/122]). In terms of functions, most of the HIA practitioners worked as consultants (59.8% [79/122]), HIA educator/lecturer/trainer (57.4% [70/122]) and HIA researcher (53.3% [65/122]). About a third (33.6% [41/122]) reported being active as HIA appraiser/reviewer.

In terms of HIA practice experience, both junior (up to five years of experience; 38.8% [47/121]) and more experienced (>10 years of experience; 37.2% [45/121]) HIA practitioners were represented. The majority of respondents with >10 years of HIA practice experience were from Europe (37.2% [45/121]).

### 3.2. HIA Practice

Almost half (46.7% [5/118]) of the HIA practitioners reported having carried out most of their HIAs in Europe (Figure 2a), where the majority of respondents were based. The second most represented region was sub-Saharan Africa (22.1% [27/118]), with 9.0% (11/122) of survey respondents originating from this region. This finding is linked to the fact that, in addition to the 11 HIA practitioners from sub-Saharan Africa, 11 HIA practitioners from Europe, four from North America, and one from East Asia and Pacific reported sub-Saharan Africa as the primary or secondary region where they have conducted HIA assignments.

The majority of practitioners (88.1% [104/118]) selected HIA of projects as their primary or secondary application field (Figure 2b). Around half of HIA practitioners also reported being experienced in HIA of policies (56.8% [67/118]) and HIA of plans (43.2% [51/118]). The least experience was reported for HIA of programmes (32.2% [38/118]) and strategies (28.8% [34/118]).

When asked about the types of health assessments conducted, a variety of different types were reported (Figure 3a). Overall, standalone HIA (79.7% [94/118]) and health in EIA (59.3% [70/118]) were the most frequently mentioned type of health assessments, followed by ESHIA (30.5% [36/118]) and environmental and social impact assessment (ESIA) (27.1% [32/118]). Specialised types of HIA, such as environmental health impact assessment (EHIA), health inequalities impact assessment, and mental well-being impact assessment, were also reported.

When asked about the decision-maker on the type of health assessment to be conducted, the HIA practitioner her-/himself (including employer) (61.5% [72/117]) and the client (47.9% [56/117]) were most frequently mentioned (Figure 3b). Under the *other* category (13.7% [16/117]), different types of public authorities were mentioned as decision-makers for the various types of health assessments. This was also dependent on the country and applicable policies of the country where it was conducted.

With regards to the preferred source documents that guide HIA practice, the World Health Organization (WHO) topic page on HIA [16] (39.5% [47/119]) and the Gothenburg consensus paper on HIA published by WHO Regional Office Europe in 1999 [8] (37.8% [45/119]) (Table 1) were reported as the main sources used. Birley’s (2011) HIA principles and practice book [11] (25.2% [30/119]), the “IAIA HIA international best practice principles” [10] (24.4% [29/119]), and the HIA guide from University of New South Wales [17] (22.7% [27/119]) were the other most commonly reported sources.

More than half of the survey respondents (57.8% [67/116]) have consulted the “IAIA HIA international best practice principles” [10], with the following reasons cited: (i) To learn about HIA in general (56.7% [38/67]); (ii) to use it as a reference for a report/publication (53.7% [36/67]); (iii) to orientate personal HIA practice (41.8% [28/67]); (iv) to use it for teaching purposes (34.3% [23/67]); (v) to share it with colleagues who wanted to learn about HIA (29.9% [20/67]); and (vi) to share it with clients who wanted to learn about HIA (11.9% [8/67]). On the question of whether the Quigley et al. [10] guidance paper needs to be revised, the majority of respondents (65.9%% [58/88]) felt that *minor revisions* should be made and more than a quarter (28.4% [25/88]) indicated that *major revisions* are needed. Few respondents (5.7% [5/88]) shared the opinion that the paper is fine as it stands, thus, not needing any revisions.

As can be seen in Table 1, the eight most frequently accessed HIA source documents have been published between 1999 and 2011. More recent source documents (2014-2016) seem to be referred to less commonly.

### 3.3. HIA Capacity Building

Out of 120 HIA practitioners, two thirds (63.3% [76/120]) have received some specific training on HIA. This proportion was highest in the practitioners from South America (100% [3/3]) and East Asia and Pacific (80.0% [12/15]), followed by Europe (60.3% [38/63]), Australia and New Zealand (62.5% [5/8]), and sub-Saharan Africa (50.0% [5/10]) (Figure 4a).

Of the 76 HIA practitioners who had received any HIA training in the past, two thirds (65.8% [50/76]) reported that the training took place at a university (Figure 4b). The IMPACT course at the University of Liverpool was the most common location for training (reported by 20 practitioners). About ten practitioners received HIA training from government sources, with five survey respondents reporting to have received training from the World Bank or the Asian Development Bank.

The areas of HIA teaching and training that the survey respondents have been engaged with are shown in Figure 5. About half of them (48.2% [39/81]) were involved in HIA teaching and training at universities. A considerable portion of the HIA practitioners (27.2% [22/81]) were delivering HIA teaching/training to government employees and/or local authorities. A few also mentioned HIA teaching and training at the level of private institutions (13.6% [11/81]) and public health institutions (12.4% [10/81]).

Two-thirds of the HIA practitioners (66.9% [81/121]) have been engaged as facilitators in HIA teaching and/or HIA training. The proportion of practitioners who have served as HIA teacher/trainer was highest in Australia and New Zealand (87.5% [7/8]), East Asia and Pacific (80.0% [12/15]), and Europe (73.0% [46/63]). Among the respondents from North America and sub-Saharan Africa, only 42.1% (8/19) and 36.4% (4/11) have ever been active as a HIA teacher/trainer, respectively.

### 3.4. Trends in HIA Practice

Survey respondents were asked to indicate the geographical locations where they have HIA experience and whether they perceive the use of HIA is increasing, stagnant, or decreasing. A total of 174 responses were obtained, showing strong geographical variation (Figure 6). In Europe (74.6% [47/63]), North America (73.9% [17/23]), and the East Asian and Pacific (55.0% [11/20]) regions, an increasing trend in HIA use was perceived by the majority of respondent HIA. Stagnant HIA practice was noticed by the majority of respondents in Latin America (71.4% [5/7]), South Asia (69.2% [9/13]); Central Asia (66.7% [2/3]), Middle East and North Africa (62.5% [5/8]), and Australia (54.6% [6/11]).

For all regions, there was always a minority of respondents who perceived a decreasing trend in HIA practice (range: 0.0-33.3%). HIA practitioners with experience in sub-Saharan Africa were almost equally divided between increasing and stagnant practice with 46.2% (12/26), and 50.0% (13/26) of responses, respectively. Across all world regions, increasing HIA use received the majority of the votes (56.9% [99/174]), followed by stagnant HIA use (35.6% [62/174]).

One hundred and fifteen survey respondents reported various main drivers for HIA practice; these can be grouped into two categories: (i) Drivers at the national level; and (ii) drivers at the international level. At the national level, the importance of policy frameworks and political interest were pointed out by half of the HIA practitioners (47.8% [55/115]). Examples received include the *Health in All Policies* approach, *Environmental Protection Acts* and France’s *Healthy Urban Planning* policies as essential mechanisms for promoting HIA practice. Another driver that was mentioned by 12 survey respondents (10.4% [12/115]) is *health and health equity advocacy*, being a promotor that can be triggered by the general public or the political level. Also, *awareness and/or interest in social determinants of health* was considered a driver for HIA that is directly connected to sustainable development and the Sustainable Development Goals (SDG) of the 2030 Agenda [28]. Furthermore, specific legal requirements were mentioned, such as the inclusion of health in EIA (12.2% [14/115]) and *indigenous peoples’* rights (0.9% [1/115]).

At the international level, integrating health in EIA was also mentioned as an enabling factor for the use of HIA (4.4% [5/115]). The European Union’s amended EIA Directive [29] was cited as an example, explicitly requiring examination of the likely significant effects of a project on human health. Furthermore, environmental and social sustainability requirements by international finance institutions (e.g., World Bank, Asian Development Bank), as well as good international industry practices promoted by the private sector (e.g., ICMM [21] and IOGP/IPIECA guidelines on HIA [22]) were reported as drivers of HIA practice (13.9% [16/115]).

In terms of barriers to HIA use, 45 of the survey respondents (39.5% [45/114]) identified a lack of technical expertise and capacity for conducting HIA as the main focus. Many expressed the need to promote HIA teaching and training, as well as the certification or accreditation of HIA practitioners. The second most reported barrier (28.9% [33/114]) was the absence of policies and legal frameworks, that specify the circumstances under which HIA is required, and the extent to which it is required. A general lack of awareness and understanding of HIA by decision-makers, as well as public health specialists, was identified as a barrier by 25 survey respondents (21.9% [25/114]). This was often brought into connection with a lack of HIA research and difficulties in demonstrating the benefits of HIA. Funding for HIA was perceived as a barrier by 17 survey respondents (14.9% [17/114]) in two ways: (i) There is often limited funding for HIA due to ambiguity in regulatory or institutional frameworks; and (ii) because conducting HIA increases the overall cost of impact assessments. Finally, uncertainty on how to integrate HIA into other forms of impact assessments such as EIA, ESHIA, or ESIA, and a lack of political will were other barriers to HIA use mentioned by respondents.

## 4. Discussion

With 122 HIA practitioners from 29 countries participating in our survey, this study provides a broadly representative sample of the existing HIA practitioner cohort, and presents a meaningful description of current HIA practice. The reasons may be due to the authors of the paper – active and well-connected HIA practitioners working all over the world – undertook a substantial outreach effort through personal networks, institutional networks (e.g., impact assessment and public health associations), and social media. It is important to note though that relative to the size of EIA or public health, the field of HIA remains small.

The proportion of HIA practitioners who participated from the different world regions seems to match well with other relatively recent descriptions of global HIA practice [6,13,30,31,32]; all pointing out that HIA practice is most developed in the European region, North America, the Asia region, and Australia. Furthermore, most of the HIA practitioners reported that, in addition to their country of origin, they are active in other countries and world regions. Our findings are consistent with other recent work, which reassures us that the experiences and views described in our study represent HIA practice globally. This is particularly relevant when considering that this paper provides important information for ongoing efforts in updating the “IAIA HIA international best practice principles” [10]. Indeed, the findings of the survey underscore the relevance of the Quigley et al. 2006 [10] paper as an important resource document for orienting HIA practice. Also, we found consensus among the large majority of the survey respondents that the “IAIA HIA international best practice principles” need to be updated, in order to better account for the diversity in current global HIA practice, as discussed in detail in the subsequent chapters.

In addition to the general direction provided by the presented research for updating the Quigley et al. 2006 [10] study, specific recommendations on how to improve distinct elements of the paper were obtained through the supplementary part of the questionnaire with a series of technical questions on the “IAIA HIA international best practice principles” [10]. The recommendations of the approximately 90 survey respondents who completed the supplementary part of the questionnaire can be summarised as follows: (i) Update the introductory part of the paper; (ii) amend the guiding principles, e.g., “democracy”; (iii) improve the HIA process section; (iv) address the diversity in methodological approaches used in HIA; (v) better distinguish between different HIA application fields; (vi) define the scale/scope of HIA; and (vii) to address the topics of health in EIA and HIA of policies in more detail. These recommendations are now considered in the revised version of the “IAIA HIA international best practice principles” [33].

### 4.1. Diversification in HIA Practice

Our findings demonstrate how HIA practice has diversified over the past two decades and that it is now recognised as an important process in protecting and promoting human health [34]. Although standalone HIA was the most frequently mentioned type of HIA conducted, the incorporation of health in other impact assessments (e.g., health in EIA, ESIA, ESHIA, or strategic environmental assessment (SEA)) has become common practice. This is in line with previous research and initiatives promoting the integration of health in other forms of impact assessments, rather than promoting standalone HIA [32,35,36,37,38]. Moreover, in the global landscape of impact assessments, specialised forms of HIA (e.g., EHIA, health inequalities impact assessment, health systems impact assessment, mental well-being impact assessment) also seem to be established types of HIA. This diversity of types of HIA can be a source of confusion for those involved in impact assessment, for example, a public or private entity proposing a new project, programme, plan, policy, or strategy (referred to as proponent), authorities, and HIA consultants, as to which type of HIA should be chosen for a given development scenario. In order to address this challenge, a rigorous scoping step needs to be applied so that the objective, scope, methodological approach, etc., of the assessment are defined early in the HIA process. Importantly, this should be done in close collaboration between the proponent, the HIA practitioners, and the regulators/competent authorities [12,33,39].

In addition to diversity in the types of HIA being practised, there is a broad range of fields of HIA application [40]. While HIA practitioners currently have the most experience in HIA of projects, a lot of experience also exists in other application fields (i.e., policies, plans, programmes, and strategies). This confirms the finding of previous research, indicating that international HIA practice has moved beyond the simple dichotomy of HIAs of projects and policies [40].

### 4.2. Trends in Global HIA Practice

In this paper, we aimed to answer whether global HIA practice is increasing, decreasing, or stagnant. An increasing trend in the use of HIA was identified for Europe, North America, and the East Asian and Pacific region. This finding is converging with the main driver of HIA practice, pointed out by the survey respondents (i.e., policy frameworks and political interest). Indeed, in the three world regions where the majority of HIA practitioners perceived HIA use to be increasing, efforts are ongoing to foster political will and strengthening policy frameworks for promoting HIA [39,41,42,43,44] or health in other forms of impact assessments [36,45,46,47]. For Australia, South Asia, and the sub-Saharan African region, HIA practitioners’ opinions were split between HIA use being stagnant or increasing. In sub-Saharan Africa, natural resource extraction projects often have to adhere to international lender standards [31], which might explain the perceived increasing trend of HIA use. On the other hand, HIA use in sub-Saharan Africa may be considered stagnant in view of the weak regulation of HIA and the absence of local HIA expertise in many countries [48]. The use in South East Asia and Oceania may also have slowed, despite having pockets of good and maturing HIA practice [49,50,51,52]. A decreasing trend in HIA use was indicated by a considerable number of HIA practitioners for Central Asia, Middle East & North Africa, and Latin America and the Caribbean. For Latin America, this confirms the findings of a recent literature review that revealed that HIA practice, and institutionalisation thereof, is still at its fledging stages in almost all countries [32]. Although perceived trends in HIA use vary globally, an overall increasing trend in the global use of HIA was observed in our study.

### 4.3. How to Further Promote HIA Practice

The most commonly mentioned barrier to the use of HIA was the paucity of technical expertise and capacity for conducting HIA. Indeed, HIA teaching and training is limited in most parts of the world [53], and with the closing of the IMPACT course in 2016 (University of Liverpool), which was a globally known HIA course. The situation may even have deteriorated in recent years. The second most-often mentioned barrier to HIA use was the lack of policies and legal frameworks that specify under what circumstances HIA is required and to what extent. This is a well-known problem in the HIA community, and previously raised in the literature [31,54]. Similar challenges apply to health in other forms of impact assessments, where the lack of formal requirements to consider health in EIA, in combination with the lack of intersectoral cooperation between the health and the environmental sectors, lead to insufficient coverage of health in EIA in terms of consideration and quality [55,56,57]. From a political and methodological perspective, the lack of effort to build capacity in HIA, coupled with the weak inclusion of health in regulatory frameworks, are limiting factors for the promotion of HIA practice. Indeed, when looking at the resource documents HIA practitioners are consulting to guiding their practice. The two references that were selected the most are: (i) the WHO homepage on HIA [12], which has not been updated for more than a decade; and (ii) the Gothenburg consensus paper [6]. The latter, published by the WHO Regional Office of Europe in 1999, comes with the following opening statement: “HIA is a rapidly developing activity. The present paper is the first in a series intended to create a common understanding of HIA.” The series has, however, been discontinued due to inactivity by the WHO headquarters in issuing guidance with global scope [14].

In order to sustain and further expand the observed upward trend in HIA practice, awareness of the HIA approach needs to be strengthened among policy-makers, financing institutions, and project proponents. Clearly, the WHO can play an important role in this by finally developing revised guidance along with reinforcing HIA awareness and capacity at the level of Ministries and agencies that are, or should be, involved in regulating and overseeing impact assessment. Furthermore, global HIA practice can capitalise on recent efforts by countries and regions, such as the European Union [47] and Lao PDR [58], which are currently strengthening the inclusion of health in impact assessment regulations. Also, the 2030 Agenda for Sustainable Development presents an opportunity for promoting health through action on the wider determinants of health [48,59,60,61]. Health is central to the three dimensions of sustainable development (social, environmental and economic) and a beneficiary of, and a contributor to, development [62]. Together, the SDGs embrace 29 health-related targets and a larger number of specified measurable indicators [63,64]. HIA cannot only be instrumental for establishing health monitoring frameworks but, most importantly, it is an essential approach for promoting intersectoral-collaboration to foster the health in-all-sectors approach, which is necessary for efficiently working towards the SDG 2030 Agenda.

Finally, there is an increasing recognition of the role that biodiversity and ecosystem services play in the relationship “healthy planet, healthy people” [65,66], and the role that impact assessments play [67]. In an outlook for the future, and additionally to providing a framework for safeguarding health in sustainable development, HIA has the potential to be contributory to the operationalisation of “planetary health” [68,69]. The systems thinking approach of planetary health with focus on the health of our civilisation and the state of our natural systems may, thus, be considered a new strategic level of operations and a new target for HIA [70]. Moreover, in future scenarios of climate change, HIA may not only identify important adverse health effects of climate change and variability, but also the opportunities and co-benefits of climate change mitigation and adaptation [71].

### 4.4. Strengths and Limitations

This study makes an important contribution to the literature on HIA, and updates previous important but now ageing descriptions of the overall state of HIA practice [12,13,31,32,72,73,74,75]. It is important to note that it was not possible to derive a specific response rate for this study due to the snowball sampling method used. As such there may have been individual HIA practitioners, or groups of practitioners, who were not invited to participate. We acknowledge that some regions (e.g., Latin America, French and Portuguese speaking countries of sub-Saharan Africa, Middle-East, and Asian regions) might be under-represented due to the survey being only conducted in English, which is the principal limitation of our study. However, given the international scope of responses, from participants at different stages of their careers, we believe this study provides meaningful insight into the current state of HIA practice internationally.

## 5. Conclusions

HIA practice has matured, grown, and diversified since the establishment of the approach more than two decades ago. For sustaining and further expanding the observed upward trend in HIA use globally, efforts are needed to address the main barriers in HIA use, namely (i) the limited technical expertise and capacity for conducting HIA; (ii) the lack of policies and legal frameworks that regulate the use of HIA or health in other forms of impact assessment; and (iii) the inadequate knowledge about HIA by decision-makers and public health specialists. The establishment of new national and international HIA teaching and training offerings seems an obvious strategy to pursue, as it addresses several of the main barriers identified, especially if courses are designed to attract participants with different disciplinary backgrounds and from different sectors. The WHO headquarters in Geneva can play an essential role in providing methodological leadership along with creating political interest in HIA, by reinforcing HIA awareness and capacity at the level of the Ministries of Health. This holds particularly true when considering that the WHO is the most important institution when it comes to orienting HIA practitioners at the global level, as demonstrated by this survey. Furthermore, IAIA is able to play a leading role in identifying good practice and in meeting the need for training and guidance, as a global network of impact assessment practitioners across different topics relating to the environment, society and health. Other networks are important in this also, for example, the European Public Health Association, the International Union for Health Promotion and Education and the Society for the Practitioners of Health Impact Assessment. Importantly, all existing and future HIA capacity building efforts and new HIA guidance documents must build on the growing recognition of the role HIA can play in operationalising the 2030 Agenda for Sustainable Development and planetary health as a new game-changing paradigm.

## Figures and Tables

**Figure 1 ijerph-17-02988-f001:**
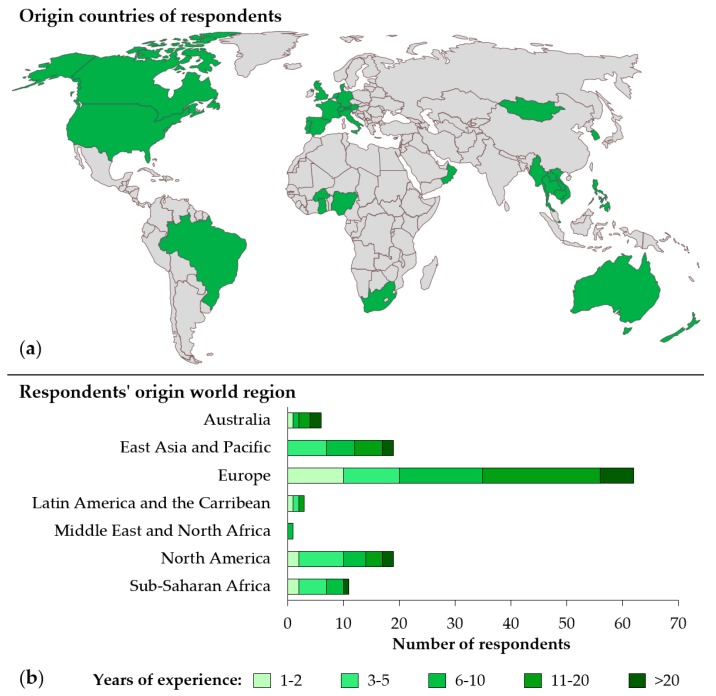
(**a**) Origin countries of respondents; and (**b**) the number of respondents per region with corresponding years of HIA practice experience.

**Figure 2 ijerph-17-02988-f002:**
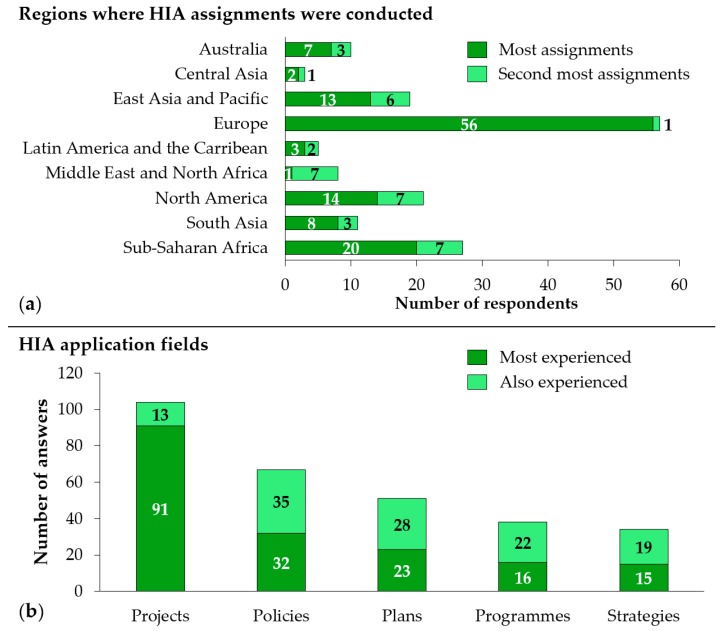
(**a**) Regions where HIA assignments were conducted; and (**b**) fields of HIA application.

**Figure 3 ijerph-17-02988-f003:**
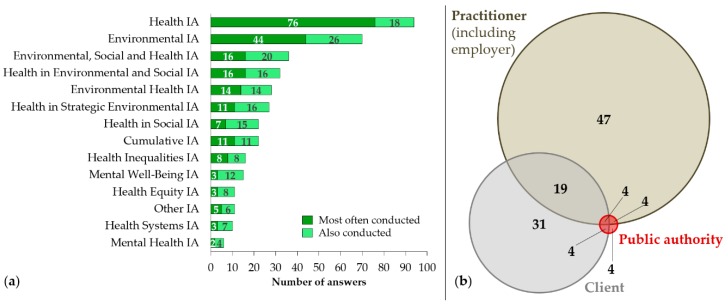
(**a**) Types of health assessments conducted; and (**b**) decision-maker on type of health assessment to conduct.

**Figure 4 ijerph-17-02988-f004:**
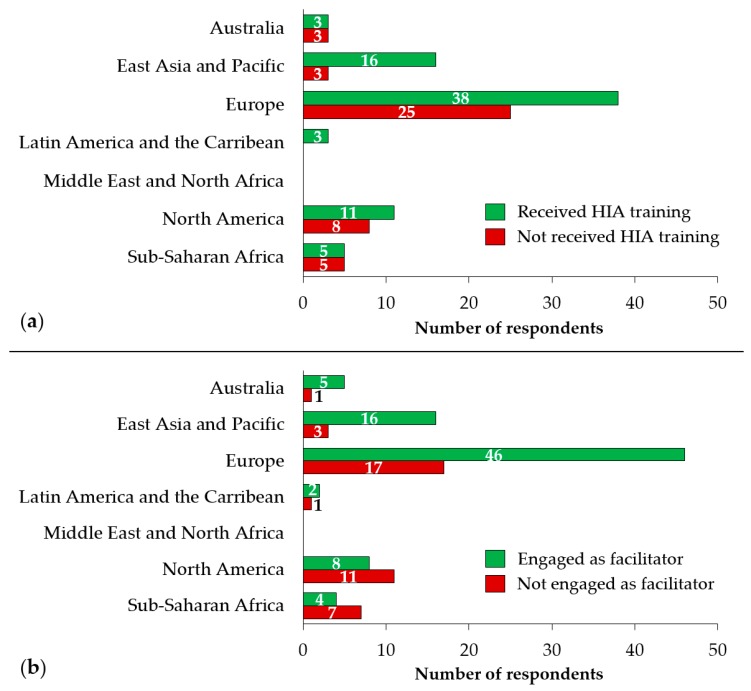
(**a**) HIA practitioners who have ever had an HIA training; and (**b**) were engaged as HIA training facilitator.

**Figure 5 ijerph-17-02988-f005:**
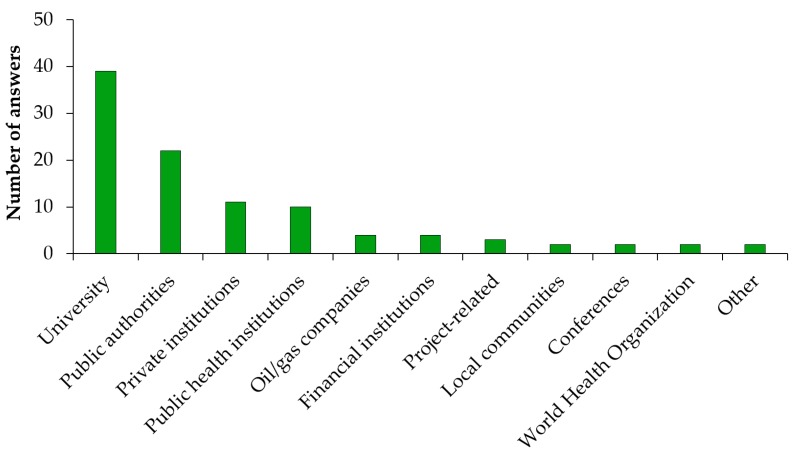
Institutions of HIA teaching and training.

**Figure 6 ijerph-17-02988-f006:**
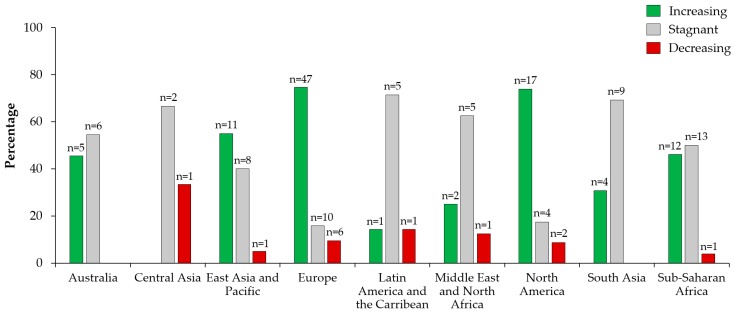
Trends in HIA use perceived by HIA practitioners with experience in the respective regions.

**Table 1 ijerph-17-02988-t001:** Preferred HIA source documents guiding HIA practice.

HIA Guidance Documents	Frequency
– World Health Organization (WHO) homepage on HIA [16]	47
– Gothenburg consensus paper: health impact assessment: main concepts and suggested approach (WHO, 1999) [8]	45
– HIA: principles and practice (Birley, 2011) [11]	30
– IAIA HIA international best practice principles (Quigley et al., 2006) [10]	29
– HIA: a practical guide (University of New South Wales (UNSW), 2007) [18]	27
– The Merseyside guidelines for HIA (Scott-Samuel, 2001) [19]	23
– International Finance Corporation (IFC) introduction to HIA (IFC, 2009) [20]	22
– Good practice guidance on HIA (International Council on Mining and Metals (ICMM), 2010) [21]	21
– A guide to HIA in the oil and gas industry (IPIECA and International Association of Oil & Gas Producers (IOGP), 2016) [22]	20
– Minimum elements and practice standards for HIA (Bhatia, 2014) [23]	18
– HIA: past achievement, current understanding, and future progress (Kemm, 2013) [12]	17
– Guidelines for the HIA of development project (Asian Development Bank (ADB), 1992) [24]	16
– IAIA FasTips: HIA (Martuzzi et al., 2014) [25]	14
– Health effects assessment tool (HEAT): an innovative guide for HIA in resource development projects (Habitat, ERM, 2010) [26]	7
– None	5
– Environmental health impact assessment in South Africa (South Africa Department of Health, 2010) [27]	2

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
