# Peer review of "Current Global Health Impact Assessment Practice"

_ijerph, 2020, doi:10.3390/ijerph17092988_

Round 1

Reviewer 1 Report

This manuscript provides a useful set of data on HIA policy and practice around the world. It is generally well-conceived, set out and explained.

I have two concerns with the paper that need to be addressed.

In the last paragraph of the Introduction, lines 71-73 the authors state that this paper is part of on-going efforts to update the IAIA HIA international best practice paper (Quigley et al. 2006). Apart from some results presented relevant to this paper, this ‘aim’ or ‘effort’ is not explicitly addressed in the Discussion. I would prefer the authors to explain their results in the context of this HIA international best practice paper.

At present the meaning of the results is directed towards the SDGs and the concept of planetary health, in 4.3. How to further promote HIA practice. This extrapolation featuresprominently in the Abstract, yet it comes despite the fact that data are not collected in the survey on these two broader issues. The extrapolation does not seem to be relevant to the aims of the paper, and it is not otherwise supported by the results gathered.

Minor matters

Paragraph starting Line 180 – can you explain the n of 88 for this paragraph?

Line 187 ‘are should’ ?

Paragraph starting line 264 – can you explain more clearly the n of 114.

Line 290 – what does thirdly refer to here?

Author Response

Reviewer #1:

This manuscript provides a useful set of data on HIA policy and practice around the world. It is generally well-conceived, set out and explained.

Response 1: We thank Reviewer #1 for the overall positive appraisal.

I have two concerns with the paper that need to be addressed.

In the last paragraph of the Introduction, lines 71-73 the authors state that this paper is part of on-going efforts to update the IAIA HIA international best practice paper (Quigley et al. 2006). Apart from some results presented relevant to this paper, this ‘aim’ or ‘effort’ is not explicitly addressed in the Discussion. I would prefer the authors to explain their results in the context of this HIA international best practice paper.

Response 2: The reviewer is raising an important point here, as we did indeed forget to elaborate on how our findings influence the on-going efforts in updating the IAIA HIA best practice paper. In order to rectify this neglect, we have made the following addition in the second paragraph of the discussion chapter: “…the experiences and views described in our study represent HIA practice globally. This is particularly relevant when considering that this paper is providing important information for the ongoing effort in updating the “IAIA HIA international best practice principles” [10]. Indeed, the findings of the survey underscore the relevance of the Quigley et al. 2006 [10] paper as an important resource document for orienting HIA practice. In addition, we found consensus among the large majority of the survey respondents that the “IAIA HIA international best practice principles” need to be updated in order to better reflect current global HIA practice.” (Lines 279-284)

At present the meaning of the results is directed towards the SDGs and the concept of planetary health, in 4.3. How to further promote HIA practice. This extrapolation features prominently in the Abstract, yet it comes despite the fact that data are not collected in the survey on these two broader issues. The extrapolation does not seem to be relevant to the aims of the paper, and it is not otherwise supported by the results gathered.

Response 3: We fully agree that the SDGs and planetary health were not specifically addressed in our survey and, therefore, might not merit to be featured in the abstract of the paper. Hence, we have replaced the sentence concerned with a concluding statement that relates more directly with the findings of the study: “… Although differences in the use of HIA were reported across world regions, an overall increasing trend in global HIA practice can be observed. In order to sustain this upward trend, efforts are needed to address the main barriers in the utilisation of HIA. The establishment of new national and international HIA teaching and training offerings seems an obvious strategy to pursue along with the strengthening of policies and legal frameworks that specify under what circumstances HIA is required and to what extent.” (Lines 32-34)

Minor matters: Paragraph starting Line 180 – can you explain the n of 88 for this paragraph?

Response 4: The n of 88 in the concerned line means that in total 88 responses were received under the question of concern in the questionnaire survey. In order to clarify what the different denominators presented in the results represent, we have added the following line at the end of the Methods section: “Since the questions asked in the online questionnaire were answered by varying numbers of respondents, the nominator (number of answers per answer category [x]), denominator (total number of responses received per question [y]) and percentage (% [x/y]) are specified for all the results presented.“ (Lines 111-114)

Line 187 ‘are should’ ?

Response 5: Many thanks for this sharp observation. We have deleted ‘are’.

Paragraph starting line 264 – can you explain more clearly the n of 114.

Response 6: See Response 4.

Line 290 – what does thirdly refer to here?

Response 7: We replaced ‘thirdly’ with “furthermore” in the revised manuscript. (Line 276)

Reviewer 2 Report

I very much appreciate the authors' work to take stock of the state of global HIA practice.  It is important to monitor the use of this assessment and evaluation tool so that necessary resources and capacity building can be provided/maintained.  The results indicate a general increase in the use and practice of HIA but there is a lack of resources, up-to-date training and funding.  Overall, I found the manuscript to be clear and compelling.  My suggestions are intended to further clarify and strengthen the bottom lines.

Specific suggestions for clarification and improvement:

Section 3.2 and Figure 3.

I would like to see the employer as decision maker presented separately rather than combined with assessor as decision maker, if possible. It seems unusual to me that the assessor would be the decision maker. 

Section 4.0 Introductory paragraphs

As described in methods the authors did undertake a significant outreach effort to increase participation. However, lacking the ability to calculate a response rate is a disadvantage that the authors gloss over too easily.  I recommend deleting the sentence in lines 283-284 ("We are confident...").  

I also recommend re-wording the sentence on lines 291-292.  It is reassuring that the findings track with other recent work and that indicates the language I would recommend.  Something along the lines of: 

That our findings are consistent with other recent work reassures us that "the experiences and views described in our study represent HIA practice globally"

Section 4.3: Given the importance of HIA as part of a Health in All Policies approach and the relevance to the SDGs I would like the authors to strengthen the Discussion with some specific recommendations.  They mention the potential beneficial impact of EU regulations - I would think that presents an opportunity for a recommendation for distribution of model regulatory language, for example.  I was also struck (but not too surprised) by the number of participants lacking formal training.  It seems that specific recommendation(s) on training and capacity building would also be welcome.

Author Response

Reviewer #2:

I very much appreciate the authors' work to take stock of the state of global HIA practice.  It is important to monitor the use of this assessment and evaluation tool so that necessary resources and capacity building can be provided/maintained.  The results indicate a general increase in the use and practice of HIA but there is a lack of resources, up-to-date training and funding. Overall, I found the manuscript to be clear and compelling.  My suggestions are intended to further clarify and strengthen the bottom lines.

Response 8: We are grateful to Reviewer #2 for appreciating our work.

Specific suggestions for clarification and improvement:

Section 3.2 and Figure 3.

I would like to see the employer as decision maker presented separately rather than combined with assessor as decision maker, if possible. It seems unusual to me that the assessor would be the decision maker.

Response 9: We agree with the reviewer that it might not be ideal that the assessor and employer are pooled as a decision-maker in the study presented. But in the questionnaire survey the answer categories available under the question “Who has generally decided about the type(s) of health assessments you have done?” were “Myself (incl. my company, my employer)”, “The client”, or “Other, please specify”. Consequently, we cannot do what the reviewer is asking for, i.e. to disentangle the assessor and employer as decision-makers about the type(s) of health assessments conducted.

In order to partly address the concern raised we have revised Figure 3b and the narrative under section 3.2 to match the wording used in the questionnaire, which provides more clarity: “When asked about the decision-maker on the type of health assessment to be conducted, the employer or HIA practitioner her-/himself (including employer) … and the client … were most frequently mentioned (Figure 3b).” (Lines 155-157)

Section 4.0 Introductory paragraphs

As described in methods the authors did undertake a significant outreach effort to increase participation. However, lacking the ability to calculate a response rate is a disadvantage that the authors gloss over too easily.  I recommend deleting the sentence in lines 283-284 ("We are confident...").

Response 10: This is a valid point made by Reviewer #2. Thus, we followed the recommendation given and deleted the sentence of concern.

I also recommend re-wording the sentence on lines 291-292.  It is reassuring that the findings track with other recent work and that indicates the language I would recommend.  Something along the lines of:

That our findings are consistent with other recent work reassures us that "the experiences and views described in our study represent HIA practice globally"

Response 11: Thanks for this recommendation. We have re-worded the sentence of concern exactly as suggested. (Lines 277-278)

Section 4.3: Given the importance of HIA as part of a Health in All Policies approach and the relevance to the SDGs I would like the authors to strengthen the Discussion with some specific recommendations.  They mention the potential beneficial impact of EU regulations - I would think that presents an opportunity for a recommendation for distribution of model regulatory language, for example.  I was also struck (but not too surprised) by the number of participants lacking formal training.  It seems that specific recommendation(s) on training and capacity building would also be welcome.

Response 12: After carefully considering this last valuable suggestion of Reviewer #2, we came to the conclusion that we prefer not to add more specific recommendations in section 4.3, justified as follows: The already quite substantial section 4.3 currently features many general recommendations but refrains from being too specific. This is because the paper per se has a global perspective and does not enter into the peculiarities of specific countries, each of which has its own realities when it comes to IA practice, regulations and capacities. Building on the example given by the reviewer, the distribution of a ‘model regulatory language’ for the inclusion of health in impact assessment might be appreciated in some countries but may be perceived as too patronising, or unfeasible by others. In order to avoid such ambiguities, we prefer to remain with the more general recommendations currently featured. This also applies to the proposed specific recommendation(s) on training and capacity building. Though, it is noteworthy that we are placing a lot of emphasis on the importance of training and capacity building, while also giving direction on how it can be promoted, not only in section 4.3. but also in the conclusion section.

Reviewer 3 Report

Congratulations on conducting this very interesting study that summarises HIA practice worldwide.

My only comments is that I would find it useful to also have information regarding the sectors (e.g. transport, mining, etc) where  the HIAs were applied. If these information is available from your survey, it would be a nice addition to the paper. However, I am not sure whether this was within the scope of your open-ended questions. 

Author Response

Reviewer #3:

Congratulations on conducting this very interesting study that summarises HIA practice worldwide.

Response 13: The positive overall appraisal of our study by Reviewer #3 is highly appreciated.

My only comments is that I would find it useful to also have information regarding the sectors (e.g. transport, mining, etc) where  the HIAs were applied. If these information is available from your survey, it would be a nice addition to the paper. However, I am not sure whether this was within the scope of your open-ended questions.

Response 14: Unfortunately we are not able to incorporate such a ‘sectoral analysis’ in our manuscript as this was not sufficiently included in our questionnaire. Regrettably, this is a missed opportunity.

Round 2

Reviewer 1 Report

The changes to the manuscript are fine but minimalistic; it will improve the paper if it is done thoroughly.

Can the authors justify the inclusions of SGDs and planetary health as keywords?

More importantly, the revised statement that the best practice principles "need to be updated" as a conclusion without saying how they need to be updated is a cursory almost meaningless change. Do we infer that the subsequent sections in the discussion are the directions for change for the best practice principles? What exactly do the survey results tell the authors about the changes required - please enumerate.

Author Response

Additional comments from Reviewer #1:

The changes to the manuscript are fine but minimalistic; it will improve the paper if it is done thoroughly.

Response 1: We thank Reviewer #1 for the additional set of comments. We did definitely not want to be minimalistic in our revisions. In the contrary, we very much appreciate all the inputs received and made an effort in addressing them in a meaningful way.

Can the authors justify the inclusions of SGDs and planetary health as keywords?

Response 2: After exposing this concern to all authors of the paper, we would still like to adhere to featuring the SDGs and planetary health as keywords. This is mainly because we see this as an opportunity to highlight that HIA and the SDG & planetary health are interlinked. Indeed, health is determined and shaped by multiple factors, and public health is working across sectors and disciplines. The health section of IAIA has been working for years on sustainable development and “planetary health”, “one health”, “healthy planet” and “healthy people”. It is the framework we are working in, which also presents external drivers that might hold the opportunity to further promote the use of HIA globally. On the other hand, HIA has been identified as an approach for operationalising planetary health and the SDG. We have elaborated on this in detail in our discussion, including external references. Hence, by featuring these terminologies as keywords, we increase the likelihood that people interested in planetary health and the SDGs identify our paper describing a field of public health practice that could be of interest to them. The other way round, HIA practitioners who have not yet reflected a lot on how HIA and planetary health and the SDGs are interlinked, might become interested to read more about it in the discussion of our paper and beyond.

More importantly, the revised statement that the best practice principles "need to be updated" as a conclusion without saying how they need to be updated is a cursory almost meaningless change. Do we infer that the subsequent sections in the discussion are the directions for change for the best practice principles? What exactly do the survey results tell the authors about the changes required - please enumerate.

Response 3: Reviewer #1 is absolutely right that we remained rather vague about how the research presented influence the ongoing revisions of the IAIA HIA best practice principles. To not address this point more specifically in the paper seems to be a weakness indeed, and we are grateful to the reviewer for insisting. In the following lines, we would like to first explain the underlying reason for the lack in specificity and then explain what was done for addressing this remaining concern.

When we reflected on how to include the larger HIA practitioner community in a potential update of the IAIA HIA best practice principles, we thought that it would be nice to conduct an online survey with global reach. When we started to develop the questionnaire tool, we realised that it would be a missed opportunity to not also investigate more broadly current global HIA practice in the frame of the online survey. What resulted was a questionnaire in two parts: The first part with various questions around HIA practice, etc. (specified in detail in the methods of the paper). The second part with specific technical questions about the different elements of the Quigley et al. 2006 paper.

In the “current global HIA practice” paper we present the findings of the first part of the questionnaire, which, in addition to provide a snapshot of global HIA practice, also gives a general direction on how the IAIA HIA should be updated (e.g. reflect diversity in practice and place emphasis on capacity building). But since the second part of the questionnaire (specific technical questions on the Quigley et al. paper) does not tell much about global health impact assessment practice, we did not include it in the results. Hence, also the discussion remained rather vague in this regard. We have now tried to address this shortcoming pointed out by Reviewer #1:

In the methods section, where details on the questionnaire tool are featured the following changes and additions were made:

“… The questionnaire tool included a mix of both closed and open-ended questions around the following topics:

  • ….
  • General questions on the “IAIA HIA international best practice principles” [10] (e.g. reasons for having consulted the paper in the past, need for revisions).
  • Specific technical questions on the “IAIA HIA international best practice principles” [10] (not reported in detail here as this is beyond the scope of the paper).” (Lines 86-89)

In the second paragraph of the discussion chapter, we do now provide detailed insights into the recommendations received under the supplementary part of the questionnaire:

“… This is particularly relevant when considering that this paper is providing important information for the ongoing effort in updating the “IAIA HIA international best practice principles” [10]. Indeed, the findings of the survey underscore the relevance of the Quigley et al. 2006 [10] paper as an important resource document for orienting HIA practice. Also, we found consensus among the large majority of the survey respondents that the “IAIA HIA international best practice principles” need to be updated in order to better account for the diversity in current global HIA practice as discussed in detail in the subsequent chapters.

In addition to the general direction provided by the presented research for updating the Quigley et al. 2006 [10] paper, specific recommendations on how to improve distinct elements of the paper were obtained through the supplementary part of the questionnaire with a series of technical questions on the “IAIA HIA international best practice principles” The recommendations of the approximately 90 survey respondents who completed the supplementary part of the questionnaire can be summarised as follows: (i) update the introductory part of the paper; (ii) amend the guiding principles, e.g. “democracy”; (iii) improve the HIA process section; (iv) address the diversity in methodological approaches used in HIA; (v) better distinguish between different HIA application fields; (vi) define the scale/scope of HIA; and (vii) to address the topics of health in EIA and HIA of policies in more detail. These recommendations are now considered in the revised version of the “IAIA HIA international best practice principles” [33].(Lines 285-299)
